# Current Trends in Diagnosis and Treatment Approach of Diabetic Retinopathy during Pregnancy: A Narrative Review

**DOI:** 10.3390/diagnostics14040369

**Published:** 2024-02-08

**Authors:** Luminioara M. Rosu, Cătălin Prodan-Bărbulescu, Anca Laura Maghiari, Elena S. Bernad, Robert L. Bernad, Roxana Iacob, Emil Robert Stoicescu, Florina Borozan, Laura Andreea Ghenciu

**Affiliations:** 1Department of Anatomy and Embryology, Victor Babeș University of Medicine and Pharmacy, E. Murgu Square, No. 2, 300041 Timisoara, Romania; rosu.luminioara@umft.ro (L.M.R.); boscu.anca@umft.ro (A.L.M.); roxana.iacob@umft.ro (R.I.); borozan.florina@umft.ro (F.B.); 2Department of Obstetrics and Gynecology, Victor Babeș University of Medicine and Pharmacy, 300041 Timisoara, Romania; bernad.elena@umft.ro; 3Clinic of Obstetrics and Gynecology, “Pius Brinzeu” County Clinical Emergency Hospital, 300723 Timisoara, Romania; 4Center for Laparoscopy, Laparoscopic Surgery and In Vitro Fertilization, Victor Babeș University of Medicine and Pharmacy, 300041 Timisoara, Romania; 5Department of Automatic Control and Applied Informatics, Politehnica University, 300223 Timisoara, Romania; robibernad11@gmail.com; 6Discipline of Radiology and Medical Imaging, Victor Babeș University of Medicine and Pharmacy, E. Murgu Square, No. 2, 300041 Timisoara, Romania; stoicescu.emil@umft.ro; 7Research Center for Pharmaco-Toxicological Evaluations, Victor Babeș University of Medicine and Pharmacy, E. Murgu Square, No. 2, 300041 Timisoara, Romania; 8Department of Functional Sciences, Victor Babeș University of Medicine and Pharmacy, E. Murgu Square, No. 2, 300041 Timisoara, Romania; bolintineanu.laura@umft.ro

**Keywords:** diabetic retinopathy, pregnancy care, gestational diabetes, diabetes during pregnancy

## Abstract

Diabetes mellitus during pregnancy and gestational diabetes are major concerns worldwide. These conditions may lead to the development of severe diabetic retinopathy during pregnancy or worsen pre-existing cases. Gestational diabetes also increases the risk of diabetes for both the mother and the fetus in the future. Understanding the prevalence, evaluating risk factors contributing to pathogenesis, and identifying treatment challenges related to diabetic retinopathy in expectant mothers are all of utmost importance. Pregnancy-related physiological changes, including those in metabolism, blood flow, immunity, and hormones, can contribute to the development or worsening of diabetic retinopathy. If left untreated, this condition may eventually result in irreversible vision loss. Treatment options such as laser therapy, intravitreal anti-vascular endothelial growth factor drugs, and intravitreal steroids pose challenges in managing these patients without endangering the developing baby and mother. This narrative review describes the management of diabetic retinopathy during pregnancy, highlights its risk factors, pathophysiology, and diagnostic methods, and offers recommendations based on findings from previous literature.

## 1. Introduction

Decreased visual acuity is considered uncommon during pregnancy, although several studies suggest otherwise. Moreover, ocular changes during pregnancy may lead to a broad range of both physiological and pathological conditions, each manifesting different symptoms and necessitating treatment. These ocular changes arise from the physiological responses to accommodate the gestational process [1,2]. While 15% of these pregnancy-induced alterations are harmless, some of these changes can impact eyesight. However, the health of the expectant mother—for example, the presence of diabetes or hypertension—has a major impact on the extent to which these ocular alterations occur. Retinal abnormalities observed in these patients can worsen during pregnancy and may be correlated with the degree of gestational diabetes mellitus (GDM).

Pregnant women may experience ocular symptoms from the retinal changes associated with either newly developed hyperglycemia or pre-existing diabetes. Globally, GDM affects 7–10% of women and is one of the most prevalent pregnancy-related disorders. It also increases the risk of developing diabetic retinopathy, which may progress to severe vision impairment [3,4]. Additionally, women with pre-diagnosed diabetes mellitus may encounter exacerbation or progression of existing diabetic retinopathy due to the heightened metabolic demands of pregnancy.

Diabetic retinopathy stands as one of the main contributors to preventable blindness in the decades II–VI in the UK and USA. This age range coincides with peak fertility for many diabetic women [5]. The onset of diabetic pathologies may occur during pregnancy, and the potential for visual impairment at this phase has significant consequences for both the mother and the fetus [6]. However, the understanding that the level of blood sugar during pregnancy is directly linked to the prevalence of congenital malformations has shifted the focus of diabetic pregnancy management towards meticulous blood sugar control. This shift has indeed led to lower rates of fetal malformations. Moreover, pregnancy-induced changes in metabolism and hormone levels can make the retinal blood vessels more susceptible to damage, increasing the likelihood of conditions such as hypertensive retinopathy. The consensus is that the mechanism underlying the progression of diabetic retinopathy during pregnancy is multifactorial, involving several contributing factors. Hyperglycemia, the course of diabetes prior to conception, the initial state of retinopathy, fast blood glucose regulation during pregnancy, co-existing systemic hypertension, preeclampsia, and alterations in retinal blood flow are important factors in this process [1,7,8]. Additionally, it has already been demonstrated that inflammatory responses and immunological phenomena may play a role in the etiology and development of DR [9,10].

Nevertheless, it is important to note that intensive glycemic control may pose risks to diabetic mothers, particularly those with established microvascular diseases like retinopathy and nephropathy. Regular eye examinations and close collaboration between healthcare providers specializing in both obstetrics and ophthalmology are crucial for ensuring the well-being of both the mother and the developing fetus. The retinal outcome of laser photocoagulation in these women and the recommended course of treatment are similar to those for other patients with diabetes. Given that postnatal regression is common and vascular proliferation can be reversed, the majority of eye specialists would typically advocate against photocoagulation treatments.

Our aim was to describe retinal changes during pregnancy related to hyperglycemia and to review the prophylaxis, diagnosis, and treatment of diabetic retinopathy.

## 2. Diabetes Mellitus and Diabetic Retinopathy during Pregnancy

The World Health Organization (WHO) and the International Federation of Gynecology and Obstetrics (FIGO) subclassify hyperglycemia during pregnancy as either gestational diabetes or diabetes in pregnancy. The disease known as gestational diabetes is typically discovered during the first 24 weeks of pregnancy [11]. Pregnant women who match the WHO criteria for diabetes in the normal population but have a personal history of diabetes or hyperglycemia that was first diagnosed during pregnancy are referred to as having diabetes in pregnancy. The majority of cases are due to gestational diabetes mellitus, with a range between 75% and 90%, with the rest being attributed to diabetes in pregnancy.

The classification of abnormalities in glucose levels first detected during pregnancy is still subject to debate. In adults, the distinction is made between diabetes mellitus and pre-diabetes—impaired glucose tolerance (IGT) and/or impaired fasting glucose (IFG). In 2013, the WHO adopted the gestational diabetes mellitus criteria proposed by the International Association of Diabetes and Pregnancy Study Groups (IADPSG), represented in Table 1.

According to these recommendations, patients with FPG  ≥  126 md/fl at any time during pregnancy and/or 2 h-PG  ≥  200 mg/dL in the OGTT are diagnosed with diabetes mellitus in pregnancy (DMP), rather than with GDM. By definition, these pregnancies are correlated with a greater risk of congenital malformations and diabetes mellitus-related complications, and these women usually would not require repeat testing postpartum to confirm the diagnosis of diabetes.

Several studies have shown that diabetic retinopathy (DR) which occurs during pregnancy usually regresses during the postpartum period. Between 57% and 62% of pregnant women with type 1 diabetes and between 17% and 28% of women with type 2 diabetes had DR at their initial evaluation [7,12,13,14,15]. When comparing the retinal changes of type I diabetes with those of type II diabetes, DR was observed to be more severe in the first case. The diagnosis of DR in pregnant women is the same as DR in non-pregnant adults (Table 2).

## 3. Risk Factors

### 3.1. Duration of Diabetes

Numerous studies have indicated that a longer duration of diabetes is associated with a higher likelihood of retinopathy progression during pregnancy [7,12,16,17]. Dibble et al. conducted a study tracking 55 insulin-dependent diabetic women throughout their pregnancies. According to their research, the duration of diabetes and the rate at which retinopathy progressed were positively correlated [18]. However, the Diabetes in Early Pregnancy (DIEP) investigation revealed that the progression of diabetic retinopathy was more pronounced in individuals with inadequate diabetes control and those who already had retinopathy at the time of conception. Notably, the research demonstrated that the duration of diabetes might not be a crucial risk factor for a significant advancement of retinopathy by two or more steps, unlike the baseline severity of retinopathy. However, the duration of diabetes does play an important role, particularly in the evolution of proliferative diabetic retinopathy.

The presence of diabetic retinopathy was found to have a strong association with the duration of diabetes. This shows the importance of acknowledging the duration of diabetes as a key factor in understanding and predicting the progression of diabetic retinopathy. This study highlights the complex relationship between diabetes control, baseline retinopathy severity, and the duration of diabetes in influencing the risk and progression of diabetic retinopathy [7].

The duration of diabetes alone is not always correlated with poor outcomes. Evidence from a study in which HbA1c was maintained at 6.6% at presentation reveals that 71% of women who had had diabetes for more than 20 years showed no development during pregnancy and either no or limited DR [14].

### 3.2. Metabolic Control

Evidence suggests that poor glycemic control has been found to be a risk factor for the development of diabetes mellitus during pregnancy in patients who had both type 1 and type 2 diabetes [7,16,17]. Wang et al. observed that the deterioration of retinopathy was obvious primarily during the first 6–12 months of strict treatment, but dramatically decreased after that [19]. The DIEP research further demonstrated that the progression rate of DR doubled in individuals whose baseline glycosylated hemoglobin exceeded 6 standard deviation (SD) above the control mean, compared to those with baseline glycosylated hemoglobin within 2 SD of the control mean [7]. Phelps et al. found that worsening of DR occurred in individuals with poor diabetes control at presentation and with either tight or intensive diabetes control in the first trimester of pregnancy [8]. In Axer-Siegel’s study, glycosylated hemoglobin levels were higher in the progressive retinopathy group compared to the nonprogressive retinopathy group, and in the DR group compared to the non-DR group, predominantly in the third trimester [20].

The Diabetes Control and Complications Trial (DCCT) compared the impact of pregnancy on both the “conventional treatment” and the “intensive treatment” groups. Pregnant women in the conventional group were switched to the intensive therapy group during pregnancy. Pregnant women in the aggressive treatment group were more likely to experience a worsening of the DR than non-pregnant women, and in the conventional group, the chance of worsening was even higher than it was in the first case. These findings highlight the association between glycemic control, pregnancy, and the progression of diabetic retinopathy.

Despite the short-term risk of DR worsening, the long-term benefits of optimizing glycemic control outweigh this concern in the general diabetic population. In pregnant women, it is advised to achieve optimal glycemic control before and as soon as possible after conception for the well-being of both the mother and the fetus, even acknowledging the potential effect on DR progression [21,22].

### 3.3. Severity of Retinopathy

Studies have indicated that the risk of visual loss is minimal in patients without pre-existing retinopathy [6]. About 12% of women who did not present retinopathy before pregnancy are likely to develop minor signs of retinopathy characterized by a few microaneurysms, but regression during the postpartum period is typically observed [23]. Soubrane et al. used fluorescein angiography to show that, in pregnant women with a history of mild diabetic retinopathy, the number of microaneurysms gradually increased during pregnancy but significantly decreased after delivery, though it did not fully return to preconception levels [24].

However, in cases where pre-existing retinopathy is more severe, there is a notable risk of proliferative changes. For instance, in the DIEP study, 29% of patients with a baseline classification of moderate retinopathy went on to develop proliferative changes while pregnant, compared to women with minimal retinopathy at baseline, of whom only 6.3% progressed to the proliferative category. This emphasizes the varying impact of pregnancy on retinopathy progression based on the severity of pre-existing retinal conditions [7].

A review of 14 studies reporting the progression of DR based on baseline levels revealed that DR occurred in 14.8% of women overall. Only one study exclusively focused on women with type 2 DM, and the rates of developing DR in this study were comparable to those in type 1 DM (11.6% and 13.1%, respectively). Women with type 2 DM in this analysis had a higher proportion without retinopathy at baseline (86.3% versus 54.2%) [25].

When examining the overall progression of proliferative diabetic retinopathy (PDR) from the absence of DR or only mild microaneurysms early in pregnancy, it was found to be rare, occurring in only three individuals (0.4%), all of whom had type 1 DM. This underscores the significance of considering baseline retinopathy levels and diabetes type when assessing the risk of DR progression during pregnancy.

### 3.4. Retinal Bloodflow

Pregnancy induces changes in the systemic vascularization, characterized by an increase in cardiac output and plasma volume, together with a reduction in peripheral resistance, resulting in an overall elevation in blood flow. Chen et al. used laser Doppler velocimetry to evaluate retinal blood flow and found that it remains stable during normal pregnancy. This study highlights the efficacy of autoregulatory mechanisms in the retinal blood vessels [26].

Diabetic patients with retinopathy experienced an observed rise in blood flow during the initial trimester of pregnancy. Nevertheless, women with diabetes who maintained a consistent retinal blood flow did not experience the development of retinopathy. This indicates that the increased blood flow in the circulatory system during early pregnancy is accompanied by compensatory mechanisms. These mechanisms are observed in both healthy women and diabetics who are able to regulate blood flow in the retina. Nevertheless, in certain diabetic women, these self-regulating systems may be compromised, leading to heightened blood circulation. The hyperdynamic circulatory state may expose the blood vessels to increased shear stress, leading to endothelial injury, especially at the capillary level [27].

It is important to note that the compensatory increase in blood flow observed in diabetic women with worsening retinopathy might be linked to local hypoxia. This could explain the observed increase, suggesting that it may be more of an epiphenomenon rather than a decline of autoregulation during pregnancy.

### 3.5. Hypertension and Preeclampsia

Hypertension represents an important risk for the development of retinopathy, and its impact becomes even more critical during pregnancy [11,17]. Rosenn et al. conducted a study tracking 154 insulin-dependent diabetic women during their pregnancies, with about a third experiencing either chronic hypertension or pregnancy-induced hypertension or a combination of both. Among those with hypertension, 55% witnessed the progression of retinopathy, compared to 25% in those without hypertension [17]. Nevertheless, a prospective study observed 51 women during pregnancy and witnessed that the blood pressure increased and the retinal arteriolar diameter decreased in diabetic women. Nonetheless, diabetic women who smoked both before and during their pregnancies did not exhibit elevated systemic blood pressure and arteriolar constriction linked to pregnancy [28].

Two studies identified a higher systolic blood pressure as a risk factor for the progression of DR. Patients with progressive DR during pregnancy exhibited higher systolic blood pressure than patients with no signs of progression [16,20].

Drug therapy for hypertension during pregnancy is most appropriate for cases of early onset and marked disease when there is a need to delay delivery. Medications such as methyldopa, beta-blockers, and vasodilators have shown some success in such situations [29]. A recent report suggests that treating women with diabetes using an angiotensin-converting enzyme (ACE) inhibitor for six months before pregnancy onset can reduce proteinuria, improve kidney function, and contribute to favorable maternal and fetal outcomes [30].

However, caution is advised regarding the use of ACE inhibitors during pregnancy, as they are known to be highly fetotoxic, resulting in complications such as hypotension, growth restriction, renal tubular dysplasia, anuria/oligohydramnios, and fetal death [31]. Despite these risks, conducting clinical studies of ACE inhibitors in DR before pregnancy onset could be valuable in assessing potential benefits while avoiding the adverse effects associated with their use during gestation.

GDM and preeclampsia share several risk factors, with pre-pregnancy obesity being a common factor. Barquiel et al. demonstrated that among patients with GDM, pre-pregnancy weight was observed as the most significant risk factor for the development of preeclampsia [32]. Furthermore, gestational weight gain and glycemia levels during pregnancy were identified as independent risk factors for preeclampsia in women with GDM. These findings are similar to those from the Hyperglycemia and Adverse Pregnancy Outcomes (HAPO) research, emphasizing the multifaceted nature of these risk factors and their impact on maternal health during pregnancy [33]. Pregnant multiparous women with pregestational DM (especially type I) should be evaluated for an elevated risk of preeclampsia. The higher frequency of risk variables (microalbuminuria, systemic hypertension, inadequate preconceptional glycemic management) is probably the cause of this elevated risk [34].

### 3.6. Other Risk Factors

A limited number of studies have found various risk factors associated with the advancement of DR. These factors comprise a younger age at which diabetes mellitus (DM) begins, the use of insulin before pregnancy in individuals with type 2 DM, lower visual clarity at the start of the study, and the existence of diabetic macular edema (DME) at the beginning of the study [21,24,35]. The studies evaluated did not consistently identify parity as a risk factor for the advancement of DR [32]. It is important to note that the evidence on these risk factors may vary across studies, and additional research is needed to further validate and understand their impact on the progression of DR during pregnancy.

A high degree of arsenic exposure may predispose to GDM, as evidenced by the correlation between GDM and arsenic levels in maternal blood, urine, and meconium identified in several investigations. Chronic arsenic exposure affects many individuals, mostly via polluted drinking water or from eating certain foods like seaweed or rice [36].

A meta-analysis shows that pregnancies with endometriosis carry a higher risk of GDM. More severe stages of endometriosis showed a notable development sequence with a notably increased risk of GDM. It is now believed that the systemic inflammation linked to endometriosis is the cause of GDM in those who have this disease. Low-grade chronic systemic inflammation and elevated insulin resistance and glucose intolerance are associated with GDM. This pathology increases the production of inflammatory effectors like leptin, interleukin-6, and tumor necrosis factor-alpha while reducing adiponectin production, which may result in insulin resistance [37].

## 4. Biomarkers in the Diagnosis of Diabetic Retinopathy

Copious research has been conducted in an attempt to find trustworthy DR biomarkers. Blood markers are the natural choice, as blood is a quickly obtained specimen with an adequate acceptability rate among the usual patient group, which is necessary for an optimum screening sample to be easily obtainable.

### 4.1. Advanced Glycation End-Products (AGE)

Numerous AGEs have been examined as possible DR biomarkers. Higher circulating levels of AGEs are generally linked, though not always, to issues related to diabetic vascular disease, such as retinopathy. Nonetheless, it is important to take into account concurrent renal illness, which typically raises AGEs. Of particular importance is carboxymethyllysine (CML), which has been shown to be higher in the blood serum of individuals suffering from DR [38]. Mishra’s results were generally consistent with previous research that discovered N-epsilon-CML(Nε-CML) to be an effective marker of DR [39]. A significant discovery was that there was a statistically significant increase in Nε-CML in NPDR as opposed to PDR. This was thought to be a sign that, in the early stages of DR, Nε-CML has pathological repercussions on retinal microvascular function. A research study that evaluated serum levels of Nε-CML in DR also revealed a correlation between high CML serum levels and proliferative retinopathy as well as clinically severe macular edema [40]. According to other studies, diabetic patients with retinopathy had higher serum levels of methylglyoxal-derived hydroimidazole than those without retinopathy [41].

Nε-CML seems to have the most potential among AGEs as a screening marker for diabetic retinopathy.

### 4.2. Vascular Endothelial Growth Factor (VEGF)

VEGF is now thought to act as both a PDR initiator and a modulator of NPDR. Since VEGF has been extensively researched and is thought to play a number of functions in the pathophysiology of DR, it seems reasonable to use it as a marker. Research has demonstrated a positive correlation between serum concentrations of VEGF and the incidence of DR [42], and a correlation with the severity of retinopathy [43]. VEGF concentrations were measured in 19 patients with NPDR and 20 patients with PDR by Jain [44], and the results were juxtaposed with those of 19 diabetics without ocular disease and 19 healthy controls. The degree of retinopathy increased substantially with VEGF. The results of Cavusoglu’s study [43] were derived from cross-sectional data examining 18 healthy controls, 34 patients with PDR, and 31 patients with NPDR. The study did not include a group of diabetics without ocular conditions as a comparison for the VEGF concentrations, which restricts its capacity to point out its potential as a screening tool. However, there was a substantial rise in VEGF levels between individuals with NPDR and PDR. According to recent studies, vitreous and aqueous VEGF levels are substantially greater in DR patients than plasma levels and are both linked to the advancement of the disease [45].

Despite the relatively small sample size of patients with ocular pathology in each cohort of the separate studies discussed above, the finding of a statistically higher serum or plasma level of VEGF with increasing disease severity was significant. This implies that VEGF might be a helpful clinical indicator of the existence and intensity of DR.

### 4.3. Anti-Inflammatory Markers

Although it is known that inflammation plays a crucial role in the pathogenesis of DR, inflammation is also closely linked to the other microvascular problems of DR, indicating that using inflammatory markers as the sole indicator of DR may not be enough.

The inflammatory response contains a variety of biomarkers, many of which are detectable in the vitreous, retina, and even the systemic circulation at altered, typically elevated, levels. In 224 subjects—out of which 24 had NDR, 144 had mild to moderate NPDR (classified as non-vision-threatening), and 57 had severe NPDR or PDR (defined as vision-threatening)—Sasongko [46] examined serum hsCRP levels. The study discovered that DR patients had a statistically significant rise in their serum hsCRP level that threatened vision. Another inflammatory analysis that has been demonstrated to rise with increasing disease severity is serum α2 antiplasmin [47]. It was noted that a recent cross-sectional study demonstrated considerably raised levels of serum α2 antiplasmin even in the early stages of ocular disease, and that there was a significant difference between DR and NDR patients. It has been demonstrated that the mean serum levels of nitric oxid (NO), soluble interleukin-2 receptor (sIL2R), IL-8, and tumor-necrosis factor-alpha (TNF-alpha) rise with the stage of DR, with PDR patients having the highest concentration [10]. Although the degree of inflammation in diabetic retinopathy is frequently thought to be minimal, several inflammatory markers have been identified as having relatively high amounts [48]. A slight association was discovered between DR and some inflammatory markers included in the panels of substances tested in several of the assessed studies [49,50].

The majority of the inflammatory markers that have been studied are not able to reliably identify DR during its first phase; instead, they may be more helpful in determining the severity of an existing illness rather than in identifying a new one.

## 5. Management of Glucose Levels and Diabetic Retinopathy

Maternal hyperglycemia during the early weeks of pregnancy has been strongly linked to a high risk of spontaneous abortions and major congenital malformations. High blood sugar levels during this critical period can have significant implications for fetal development, leading to adverse outcomes. Therefore, maintaining an optimal glycemic level is crucial to lowering the risk of such complications and supporting the healthy development of the fetus [51]. The recommendations for pregnancies complicated by diabetes include diagnostic and therapeutic actions known or believed to have a favorable impact on both maternal and perinatal outcomes. We have reviewed studies and guidelines regarding both pregnant and non-pregnant patients with diabetes to optimize management during pregnancy and control complications [52,53,54,55,56,57,58].

### 5.1. Evaluation during Preconception

Preconception care is essential to ensure optimal maternal health and reduce the possibility of complications for both the mother and the developing fetus. Education and awareness about the importance of good glucose control before pregnancy are crucial for women with diabetes during peak-fertility years, as pregnancy can seriously affect the management of DM and can be associated with the progression of DR [59,60,61]. Population-based data consistently indicate higher rates of congenital malformations and increased perinatal morbidity and mortality in pregnancies affected by diabetes [62]. During pregnancy, placental hormones (estrogen, cortisol, human placental lactogen) [63], growth factors (IGF-1, IGF-2, IGFBP1-1, placental growth hormone) [64,65,66,67], cytokines (IL-6, IL-8) [68], and other biomolecules [69] contribute to a continuous increase in insulin resistance. This necessitates intensive medical therapy and often requires adjusted insulin administration to prevent threatening hyperglycemia for the fetus. In reaction to the stress of coexisting illnesses or drugs taken for the treatment of obstetrical issues, the elevated insulin resistance may also cause ketoacidosis [70].

In addition, pregnant women with type 1 DM are at a higher risk of experiencing insulin-induced hypoglycemia, which can be very dangerous for them. Women who have type 2 DM often start pregnancy with significant insulin resistance and obesity, which adds to the difficulty of obtaining optimal management of blood sugar levels.

It is well known that achieving and maintaining optimal glycemic levels prior to and throughout pregnancy can significantly lower the risks associated with various complications. This includes minimizing the risks of congenital malformations, perinatal morbidity, and mortality [71]. The importance of preconception management is emphasized in guidelines for managing diabetes in pregnancy, such as those provided by the American Diabetes Association (ADA) [72]. However, an editorial by Murphy raises the concern that pre-pregnancy counselling tends to attract well-educated and socioeconomically advantaged women, who typically have a lower risk of adverse pregnancy outcomes. On the other hand, women at a higher risk, who may benefit most from preconception care, are less likely to seek or have access to these services [73]. The study by Skajaa, involving 380 women with type 1 diabetes and a total of 530 pregnancies, investigated the impact of pre-pregnancy HbA1c levels on HbA1c levels throughout pregnancy. The findings of the study revealed that higher pre-pregnancy HbA1c was a predictive factor for poor glycemic control throughout the entire pregnancy [74]. Maintaining blood glucose and HbA1c levels within a target range before pregnancy is crucial to reduce the risk of complications [32,75,76]. The comprehensive evaluation for women with diabetes considering pregnancy should include a thorough review of their medical history, including previous pregnancies and comorbidities [77]. This entails evaluating conditions like dyslipidemias, heart disease risk factors, hypertension, albuminuria, peripheral vascular disease, neuropathies, knowledge of hypoglycemia, severe episodes of hypoglycemia, intestinal symptoms, celiac disease, thyroid pathologies, and infections. Furthermore, the evaluation should cover the individual’s history of diabetes education, treatment, and past and present levels of glycemic control. The physical examination should extend beyond appropriate obstetrical examination to include specific assessments (orthostatic blood pressure, thyroid palpation, auscultation for carotid and femoral bruits, and checking for the presence or absence of Achilles reflexes).

Accurately determining the level of DR before conception is crucial, and many guidelines strongly recommend a preconception ophthalmic examination. This examination can be conducted through dilated fundus examination or fundus photography screening. In the UK, it is further advised that annual controls occur after the first preconception consult until pregnancy occurs [78].

Consistent with recommendations for the general diabetic population, if severe NPDR, PDR, or diabetic macular edema is present during an examination, referral to an ophthalmologist is recommended. This referral ensures that any necessary laser photocoagulation is applied before pregnancy. Existing guidelines do not provide recommendations for the use of intravitreal injections for the treatment of DME before pregnancy.

### 5.2. Pregnancy Care

In women without comorbidities before pregnancy, GDM can develop when the insulin secretory capacity becomes insufficient to counteract the reduced action of insulin caused by hormonal production from the placenta as pregnancy advances. Although there is not universal agreement on the best procedure and cutoff points for identifying and diagnosing GDM, most countries and guidelines agree that the diagnosis should involve a 75 g OGTT performed between 24 and 28 weeks of pregnancy. The approach to screening or testing for GDM varies, with some countries recommending testing for all pregnant women, while others focus on testing those with specific risk factors for GDM. Risk factors for GDM include a high body-mass index (BMI), advanced maternal age, a previous history of GDM, a family history of type 2 DM, and ethnic background. These risk factors play a crucial role in identifying individuals who may benefit from screening or testing for GDM during pregnancy [74]. A recent publication has indicated that each individual value obtained from the 2-h OGTT is correlated with a total score for genetic vulnerability. This genetic risk score is adjusted for both fasting venous plasma glucose and the risk of type 2 DM. This suggests that the genetic predisposition to elevated glucose levels and the risk of developing type 2 diabetes may be reflected in the results of the 2-h OGTT [32]. There is a lack of consensus on the specific glucose levels that should be considered diagnostic for GDM in the OGTT. Different countries and medical societies may have varying criteria for diagnosing GDM, leading to a lack of uniformity in the diagnostic thresholds.

Generally, lifestyle changes including maintaining a healthy weight, eating a balanced diet, getting regular exercise, and checking blood sugar levels are necessary for the management of GDM. Many women with GDM can effectively control their blood glucose levels through these measures [79]. However, some individuals may require medication, with insulin being the most common, and occasionally metformin, to manage high glucose levels. Large-sized pilot studies have demonstrated that treating GDM leads to improved pregnancy outcomes. Overall, the detection and treatment of GDM have been shown to reduce the risk of serious perinatal complications by 67%, decrease the incidence of macrosomia by 50%, lower the risk of shoulder dystocia by more than 50%, reduce the rate of cesarean section by 20%, and decrease the risk of preeclampsia by 30–50% [80]. Achieving glycemic control during the first trimester of pregnancy and maintaining it throughout gestation is linked to the lowest incidence of maternal, fetal, and neonatal disorders. It is crucial to develop or adjust the management plan to aim for near-normal glycemia while minimizing the risk of significant hypoglycemia. The recommended glycemic goals during pregnancy are as follows: premeal, bedtime, and overnight glucose: 60–99 mg/dL; peak postprandial glucose: 100–129 mg/dL; mean daily glucose: <110 mg/dL; HbA1c: <6.0%. These specific targets are designed to minimize the risk of complications and ensure the best possible outcomes during pregnancy [81,82].

The National Institute for Health and Clinical Excellence (NICE) established guidelines for the management of diabetes during pregnancy in 2008 (Table 3) [83]. The outcomes of several studies revealed that the majority of women were managed in accordance with NICE guidelines. However, the remaining individuals did not meet the specified targets, primarily due to the non-attendance of appointments. This underscores the importance of addressing barriers to attendance and implementing strategies to ensure that a higher proportion of pregnant women with diabetes receive the recommended screening in alignment with NICE guidelines.

#### 5.2.1. Clinical Investigations

An eye examination in the first trimester of pregnancy is recommended by all guidelines. Color fundus photography has been a valuable tool for documenting the severity of DR. The Early Treatment Diabetic Retinopathy Study (ETDRS) group initially established that stereoscopic color fundus photography in seven standard fields (30°) is the most reliable method for diagnosing DR [84]. Recently, wide-field fundus photography has been utilized and has demonstrated a strong association with the seven conventional color mydriatic fields [85]. Optical coherence tomography (OCT) is a highly effective method for evaluating the extent and intensity of diabetic macular edema (DME) and identifying retinal structural alterations caused by ischemia [86,87]. Optical OCT has emerged as the universally accepted benchmark for the diagnosis and ongoing assessment of DME. Its reliability and repeatability contribute to a greater level of accuracy compared to fundus photography in the diagnosis of DME [88,89]. The combination of color fundus photography and OCT provides comprehensive information for assessing and managing diabetic retinopathy, allowing healthcare professionals to monitor both the vascular changes associated with DR and the structural alterations indicative of DME with a high degree of precision.

Fluorescein angiography (FA) and indocyanine green angiography (ICG) are diagnostic imaging techniques used to visualize blood vessels and are of utmost importance in the diagnosis of diabetic retinopathy. Studies have indicated that when FA is performed before the 15th week of pregnancy, it does not induce adverse effects. Some reports even suggest minimal fetal abnormalities with no major complications. Fluorescein has been classified as category 3 by the United States Food and Drug Administration (FDA) and as category 2B by the Therapeutics and Goods Administration (TGA) in Australia, in relation to its safety during pregnancy. ICG, unlike fluorescein, does not pass through the placenta during any stage of pregnancy. It is also not seen in fetal blood or the umbilical cord shortly after birth. The US FDA categorizes ICG under category C, indicating that there may be risks associated with its use during pregnancy, and its benefits should be weighed against potential risks. While FA has been used in early pregnancy without reported major complications, ICG is considered safer during pregnancy, as it does not cross the placenta. However, any decision to use these angiography techniques during pregnancy should be carefully considered, weighing the potential benefits against the associated risks, and consultation with healthcare professionals is crucial for making informed decisions [90,91].

In a study conducted by Sunness et al., it was found that 58% of those identified with PDR during the first stage of pregnancy and treated with lasers experienced progression. In contrast, only 29% of those diagnosed and treated before pregnancy exhibited progression [23]. A study conducted by Hercules et al. found that thorough laser therapy resulted in a reduction of new blood vessel formation in 63% of women both before and after giving birth. However, this treatment did not have a significant impact on improving visual acuity [92]. Best et al. and Agardh et al. advocated for the use of laser therapy in PDR to facilitate regression, notwithstanding worries about macular edema [93,94]. Chan et al. recommended prompt laser intervention for pre-proliferative DR during pregnancy rather than waiting for natural regression, as postponing therapy could result in a more unfavorable prognosis [95].

#### 5.2.2. Pharmacological Treatment

Pharmacological therapy is one of the alternatives for treating GDM when lifestyle changes are not enough to manage glucose levels. Insulin, metformin, and glyburide are the three medications that are often considered for the treatment of GDM (Table 4). The recommended course of treatment for hyperglycemia in GDM is the administration of insulin. Glyburide and metformin are not thought of as first-line treatments, because they both reach the fetus through the placenta [96].

#### 5.2.3. Laser Treatment

In cases where a pregnant woman initially presents with mild to moderate diabetic macular edema, it may be reasonable to recommend close observation with a focus on achieving and maintaining optimal blood glucose control. A report from Denmark highlighted two diabetic patients in the early stages of their pregnancies who presented with macular edema located between 500 and 1500 μm from the fovea. Both patients experienced improvement in their condition with good blood glucose control, and as a result, they did not require further intervention [21]. While observation is a reasonable option for pregnant patients with mild to moderate diabetic macular edema, it is crucial to monitor these patients more closely than one would a non-pregnant adult. If DME does not show improvement after a period of observation, the first-line treatment option is laser therapy. ETDRS found that using grid or focused laser treatment for clinically significant macular edema is successful in reducing further visual impairment [97]. According to a study conducted in Copenhagen, two pregnant women diagnosed with type 1 DM and macular edema had targeted laser treatment and did not need any additional medical intervention throughout their pregnancies [15]. In cases where foveal involvement makes the safe use of conventional laser treatment challenging, alternative approaches such as subthreshold MicroPulse or Endpoint Management may be considered. Researchers from Italy demonstrated significant short-term improvement in DME and vision after treatment with MicroPulse laser therapy [98]. These alternative methods can be valuable when conventional laser treatments are not suitable or pose potential risks, especially in the context of pregnancy.

Panretinal photocoagulation (PRP) is considered a safe and effective treatment choice in pregnant women with DR. It remains a crucial intervention for preventing the progression of the disease and has been established as a mainstay in managing diabetic retinopathy during pregnancy. The timing of PRP treatment in pregnant patients is an important consideration. Recommendations suggest that treatment with PRP may be warranted at an earlier stage in pregnant patients, particularly when the level of DR reaches the severity of severe nonproliferative DR or worse [83,99].

#### 5.2.4. Intravitreal Steroids in DME

The use of intravitreal steroids during pregnancy is a topic with limited literature evidence, and information is primarily derived from smaller studies. While there may be reports and studies indicating the use of intravitreal steroids and assessing their safety profiles at various stages of pregnancy, the overall body of evidence is not as extensive as for some other interventions. It is crucial to acknowledge the ethical challenges associated with conducting comprehensive studies on pregnant women, which can limit the availability of large-scale, high-quality evidence. The safety of any medical intervention during pregnancy is a critical concern due to potential effects on both the mother and the developing fetus. Given the limited evidence, decisions regarding the use of intravitreal steroids during pregnancy should be made on a case-by-case basis, carefully weighing the possible advantages against the known or potential risks [100,101,102,103].

It is worth noting that certain studies have linked the use of systemic corticosteroids during the first trimester of pregnancy with oral clefts [104], and topical corticosteroids have been associated with small for gestational age (SGA) in some research [105]. However, when it comes to intravitreal triamcinolone, studies have demonstrated minimal systemic absorption of the drug [106].

#### 5.2.5. Intravitreal Anti-VEGF Substances

The use of anti-VEGF therapy during pregnancy is generally considered a last resort, primarily due to the limited availability of long-term safety data for this type of treatment in pregnant individuals. Anti-VEGF drugs, such as bevacizumab, ranibizumab, and aflibercept, are commonly used to treat various eye conditions, including certain retinal diseases. Pregnant women are often cautious about using medications that have not been extensively studied for their safety during pregnancy, as the potential impact on the developing fetus is a significant concern. Therefore, if anti-VEGF therapy is deemed necessary for a pregnant woman, it is typically considered only when other treatment options are not viable or effective. Furthermore, if anti-VEGF therapy is initiated during pregnancy, it is generally preferred to administer the treatment in the later stages, particularly within the third trimester. This timing is based on the principle of minimizing potential risks to fetal development during the critical early stages of pregnancy.

In a patient with foveal-involving diabetic macular edema and a contraindication to steroids, the use of anti-VEGF therapy could be considered. The Diabetic Retinopathy Clinical Research Network study has shown that anti-VEGF therapies are highly effective in the treatment of DME. This study compared the outcomes of anti-VEGF therapy combined with laser treatment versus laser treatment alone. The results demonstrated the efficacy of anti-VEGF therapies in improving the outcomes of DME [107].

The decision regarding the choice of anti-VEGF therapy is often influenced by the drug’s half-life in the plasma. Bevacizumab has a longer half-life and has been observed to remain in the plasma for an extended duration. An intravitreal dose of bevacizumab has been shown to reduce plasma VEGF levels for at least one month. In contrast, ranibizumab has a shorter half-life and undergoes rapid plasma clearance. Due to its shorter duration in the bloodstream, ranibizumab is considered a possible drug of choice in pregnant women and also in individuals who are planning to become pregnant shortly after receiving the anti-VEGF injection [108,109].

#### 5.2.6. Delivery

It is not necessary to see DR as a restriction to vaginal birth. In women with uncontrolled PDR, vaginal birth has been linked to vitreous and retinal hemorrhages caused by the Valsalva maneuver. To aid controlled pushing, the ADA advises using epidural anesthetic with assisted delivery [110]. Poor perinatal outcomes have been linked to prolonged second stages of labor. Other methods (forceps/vacuum extraction) should be taken into consideration, as they may help shorten the second stage of labor and lessen the necessity for a second-stage cesarean delivery [111].

### 5.3. Postpartum Care

Individuals with GDM face a high risk of chronic complications, which may include impaired glucose metabolism, diabetes, cardiovascular disease, and obesity. A study conducted by Catalano et al. demonstrated a consistent decrease of 50–60% in insulin sensitivity as gestation progresses, affecting both women with normal glucose tolerance and those with GDM [112]. Notably, individuals with GDM exhibit lower insulin sensitivity in late pregnancy compared to others with normal glucose tolerance. This decline in insulin sensitivity is primarily a reflection of pre-existing reduced insulin sensitivity before pregnancy. Given the long-term risks associated with GDM, it is crucial to implement a monitoring schedule for women with a history of GDM. Monitoring and managing their health, particularly in terms of glucose metabolism and cardiovascular health, can help lower the risk of complications and contribute to better long-term outcomes. Regular follow-up assessments, lifestyle interventions, and close collaboration with healthcare professionals are essential components of a comprehensive post-GDM care plan.

For women diagnosed with pre-proliferative DR during pregnancy, it is advisable to provide counseling for regular ophthalmologic check-ups for a minimum of 6 months post-delivery and, usually, up to 1 year postpartum. This follow-up care is essential for monitoring the progression of diabetic retinopathy and ensuring intervention if needed [62,113]. In the DCCT study, it was found that the increased risk of DR progression during pregnancy continued for 1 year after giving birth. Several women necessitated laser photocoagulation for a duration of 12 months following childbirth [2]. According to another study, the progression of DR was found to be more probable 4 months after birth rather than during pregnancy. This phenomenon can be related to the successful management of blood sugar levels throughout pregnancy, which then decline in the period after giving birth [114]. A dilated fundoscopy should be considered 1–2 months after childbirth for women who had treated or untreated DME and those with mild, moderate, or severe nonproliferative DR during pregnancy. This follow-up examination is recommended to assess the retinal status postpartum and should continue to be performed until 12 months after delivery [115]. Recommendations for the stages of preconception, during pregnancy, and postnatal are listed in Table 5 [62].

Despite an increased short-term risk of DR progression during pregnancy, it has been observed that pregnancy does not elevate the risk of DR. Following an average 6.5 years of check-ups in the DCCT, DR levels were found to be similar between those who had gone through at least one pregnancy and those who had not. Other studies have corroborated these findings, indicating that the risk of progress to PDR two years after delivery and the need for laser treatment at the mark of 5 and 10 years after delivery did not elevate with pregnancy [2,35,116]. Acknowledging the exact mechanisms through which risk factors can lead to the onset and progression of DR in pregnancy can visibly improve the screening and treatment strategy [117].

## 6. Conclusions

The progression of DR during pregnancy is influenced by several factors, including the severity of retinopathy at conception, the effectiveness of treatment, the duration of diabetes, metabolic control before pregnancy, and the presence of additional vascular damage such as pre-existing or concurrent hypertensive disorders. A precise identification of risk factors and subsequent effective management of diabetes reduce the likelihood that the retinopathy will progress. For women newly diagnosed with diabetes during pregnancy, an ophthalmology consult with fundoscopy is recommended, aiming to enhance early detection and management of potential ocular complications, thus safeguarding maternal and fetal eye health. For individuals with minimal retinopathy at the start of pregnancy, the risks of visual loss are minimal, and a fundus examination every 3 months is typically sufficient. For those with moderate background retinopathy, more frequent monitoring is recommended, with fundoscopy performed at each obstetric visit (usually every 4 to 6 weeks). If there is evidence of advancement, it is recommended to conduct examinations of the patient at intervals of two weeks in order to identify any high-risk characteristics. If there are indications of high-risk retinal changes, it is advisable to promptly perform laser photocoagulation and monitor the procedure using fundoscopy. For women with severe DR, laser photocoagulation should ideally be performed before pregnancy or promptly when important fundus changes appear. This comprehensive approach to monitoring and intervention is aimed at minimizing the risk of visual complications in pregnant individuals with DR.

## Figures and Tables

**Table 1 diagnostics-14-00369-t001:** OGTT criteria for diagnosis of Diabetes Mellitus (WHO, FIGO, IADPSG).

OGTT	Normal	GDM	DMP
FPG	<92 mg/dL (5.1 mmol/L)	92–125 mg/dL (5.1–6.9 mmol/L)	≥126 mg/dL (7.0 mmol/L)
1-h	<180 mg/dL (10 mmol/L)	>180 mg/dL (≥10 mmol/L)	
2-h	<153 mg/dL (8.5 mmol/L)	153 to 199 mg/dL (8.5–11.0 mmol/L)	≥200 mg/dL (11.1 mmol/L)

Abbreviations: OGTT—oral glucose tolerance test, FPG—fasting plasma glucose, GDM—gestational diabetes mellitus, DMP—diabetes mellitus in pregnancy.

**Table 2 diagnostics-14-00369-t002:** Diabetic Retinopathy grading.

DR Grade		Retinal Findings
non-proliferative	mild	microaneurysms only
	moderate	minimum one hemorrhage or microaneurysm and/or one of the following: retinal hemorrhages hard exudates cotton wool spots venous beading
	severe	any of the following but no characteristics of PDR (4-2-1 rule): >20 intraretinal hemorrhages in each of the four quadrants definite venous beading in at least two quadrants IRMA in at least one quadrant
proliferative		one of the following: neovascularization vitreous/preretinal hemorrhage

Abbreviations: DR—diabetic retinopathy; IRMA—intraretinal microvascular abnormality; PDR—proliferative diabetic retinopathy.

**Table 3 diagnostics-14-00369-t003:** NICE guidelines: retinal assessment during pregnancy.

Pregnant women who have pre-existing DM should go through dilated fundoscopy following their first antenatal consult. If the initial examination is normal, another assessment should be done at 28 weeks.
2. If DR is detected, it is recommended to perform an additional retinal examination between 16 and 20 weeks.
3. Women who are diagnosed with pre-proliferative DR while pregnant should undergo ophthalmological monitoring for at least 6 months after giving birth.
4. DR should not be considered a reason to delay the prompt improvement of glycemic control in women with high HbA1c levels during the first stage of pregnancy.
5. Diabetic retinopathy should not be viewed as a reason to avoid vaginal birth.

**Table 4 diagnostics-14-00369-t004:** Guidelines for pharmacological therapy in gestational diabetes mellitus.

Line of Treatment	Drug	Route of Administration	Dose	Observations
I	Insulin	Subcutaneous	Individualized based on glycemia levels	Preferred medication Multiple injections/pump
II	Metformin	Oral	500 mg 1×/2× per day	Requires monitoring (placental cross)
III	Glyburide	Oral	2.5 mg 1× per day	Many disadvantages, such as macrosomia and neonatal hypoglycemia

**Table 5 diagnostics-14-00369-t005:** Recommendations for the stages of pregnancy.

Phase	Recommendations	Rationale
Preconception	Dilated eye examination (ophtalmoscopy, biomicroscopy). Counseling on DR risk. Optimize glycemic and blood pressure control. Prompt referral for macular edema, severe NPDR, or PDR. Glycemic levels should be as close as possible to normal level at least 6 months before the attempt of conceiving.	Evaluate and manage risk factors before pregnancy to ensure optimal control and early intervention.
Pregnancy	Dilated fundoscopy in the first trimester, close monitoring. Frequency based on DR severity. Laser therapy for high-risk cases. Consider assisted delivery in untreated PDR.	Regular eye exams during pregnancy, laser therapy for high risk, and cautious delivery planning.
Postnatal	Continued DR assessment. Laser therapy may be needed postpartum.	Ongoing postpartum assessment with potential need for laser therapy.

## Data Availability

No new data created.

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
