# Peer review of "Current Trends in Diagnosis and Treatment Approach of Diabetic Retinopathy during Pregnancy: A Narrative Review"

_diagnostics, 2024, doi:10.3390/diagnostics14040369_

Round 1

Reviewer 1 Report

Comments and Suggestions for Authors

Your article is about Diabetes and diabetic retinopathy an increasing pathology giving the fact that GDM is now being diagnosed more frequently due to OGTT screening test.

There are some issues addressed below

line 244-245 - Are you sure that 115 and 105 are the thresholds? because these values are not considered high blood pressure during pregnancy. In your ref nr 16 the systolic blood pressure at booking of 122 was not associated with progression of retinopathy. Please clarify.

In Discussion section a recommendation regarding mode of delivery (CS vs vaginal delivery) in patients with DR should be inserted

In conclusion section a recommendation for a ophthalmology review should be inserted for every newly diagnosed diabetes during pregnancy.

Author Response

Dear reviewer, 

Thank you so much for your interest in our manuscript and for accepting this task. We are glad that you find it interesting and tried our best to improve the manuscript thanks to your remarks which I will address below: 

  1. line 244-245 - Are you sure that 115 and 105 are the thresholds? because these values are not considered high blood pressure during pregnancy. In your ref nr 16 the systolic blood pressure at booking of 122 was not associated with progression of retinopathy. Please clarify.

Yes, the other article stated these BP values. I will paste a part of the results described: "The blood pressure measurements were normal in all
patients. However, during the second trimester, systolic
blood pressure was higher than 115 mmHg in 50% of the
P group, but lower than 105 mmHg in 55% of the NP
group (P < 0.001, chi-square test). During the third trimester, 65% of patients in the P group had systolic blood
pressure higher than 115 mmHg, whereas 50% of the NP
group had systolic blood pressure lower than 105 mmHg
(P = 0.06, chi-square test)." Nevertheless, the first reference stated higher BP values, therefore we changed the sentence and left out the exact values. 

2. In Discussion section a recommendation regarding mode of delivery (CS vs vaginal delivery) in patients with DR should be inserted

We added a paragraph about delivery in patients with DR. There is also a short recommendation about delivery in Table 4. 

3.In conclusion section a recommendation for a ophthalmology review should be inserted for every newly diagnosed diabetes during pregnancy.

We also added information about check-ups in newly diagnosed diabetes.

Once again, we thank you for your time and interest.

Assist prof Dr Ghenciu Laura

Reviewer 2 Report

Comments and Suggestions for Authors

Dear Authors,

thank you for giving me the opportunity to review this narrative review on diabetic retinophaty during pregnancy. DR is a rare complication of gestational and pregestational diabetes and I believe that your narrative review provides important suggestions on how to manage this clinical entity and may be of help in the daily practice.

I don't have any particular issue to raise, but I would only recommend to go through your article because there are some typos that needs to be corrected. Other than this I think that the paper can be considered for publication.  

Comments on the Quality of English Language

The quality of English language is good but there are only minor typos to be corrected. 

Author Response

Dear reviewer,

Thank you for your time and interest. We value your opinion and we are very excited that you find the topic and the article interesting.

We have made several changes to improve the manuscript and make it more readable. We have also made a full grammar and spelling check-up.

Once again, we thank you so much for reviewing our manuscript.

Asist prof Dr Ghenciu Laura